



# Mesoscale variations in the assemblage structure and trophodynamics of mesozooplankton communities of the Adriatic basin (Mediterranean Sea)

Emanuela Fanelli[1,2*], Samuele Menicucci[1], Sara Malavolti[3], Andrea De Felice[3], Iole Leonori[3]

[1] Department of Life and Environmental Sciences, Polytechnic University of Marche, Via Brecce Bianche 60131, Ancona (Italy)

[2] Stazione Zoologica Anton Dohrn, Villa Comunale, Naples (Italy)

[3] CNR IRBIM National Research Council - Institute of Marine Biological Resources and Biotechnologies, SS Ancona, Largo Fiera della Pesca, 1 - 60125, Ancona (Italy)

*Correspondence to*: Emanuela Fanelli (e.fanelli@univpm.it)

**Abstract.** Zooplankton are critical to the functioning of ocean food webs because of their utter abundance and vital ecosystem roles. Zooplankton communities are highly diverse and thus perform a variety of ecosystem functions, thus changes in their community or food web structure may provide evidence of ecosystem alteration. Assemblage structure and trophodynamics of mesozooplantkon communities were examined across the Adriatic basin, the northernmost and most productive basin of the Mediterranean Sea. Samples were collected in June-July 2019 along coast-offshore transects covering the whole western Adriatic side, consistently environmental variables were also recorded. Results showed a clear separation between samples from the northern-central Adriatic and the southern ones, with a further segregation, although less clear, of inshore *vs*. off-shore stations, the latter mostly dominated in the central and southern stations by gelatinous plankton. Such patterns were mainly driven by chlorophyll-*a* concentration (as a proxy of primary production) for northern-central stations, *i.e.* closer to the Po river input, and by temperature and salinity, for southern ones, with the DistLM model explaining 46% of total variance. The analysis of stable isotopes of nitrogen and carbon allowed to identify a complex food web characterized by 3 trophic levels from herbivores to carnivores, passing through the mixed feeding behavior of omnivores, shifting from phytoplankton/detritus ingestion to microzooplankton. Trophic structure also spatially varied according to sub-area, with the northern-central sub-areas differing from each other and from the southern stations. Our results highlighted the importance of environmental variables as drivers of zooplanktonic communities and the complex structure of their food webs. Disentangling and considering such complexity is crucial to generate realistic predictions on the functioning of aquatic ecosystems, especially in high productive and, at the same time, overexploited area such as the Adriatic Sea.

***Key-words:*** *mesozooplankton, community composition, environmental drivers, food webs, stable isotopes, Adriatic Sea*



## 1 Introduction

In an oligotrophic system, such as the Mediterranean Sea, coastal productivity largely depends on inputs from rivers and areas of high productivity are mainly restricted to waters close to major freshwater inputs (D'Ortenzio and Ribera d'Alcalà, 2009, Ludwig et al., 2009). Here, the Adriatic basin represent an anomaly, with the northern Adriatic being one of the most productive Mediterranean areas. While the northern part is a shallow sub-basin, characterised by inputs of several rivers, with the Po representing the major buoyancy input with an annual mean discharge rate of 1500~1700 $m^3s^{-1}$, and accounting for about one third of the total riverine freshwater input in the Adriatic (Raicich, 1996), the southern part is characterized by highly saline and oligotrophic waters (Franco and Michelato, 1992; Boicourt et al., 1999). Thus, a trophic gradient, decreasing from northwest to southeast, is typically observed in the basin, in which the nutrient-rich waters coming from the rivers are mainly spread southward and eastward from the Italian coast (Bernardi Aubry et al., 2006; Solidoro et al., 2009). Such differences may be reflected in the population dynamics of the marine biotic components (Revelante and Gilmartin 1977; Simonini et al. 2004; Hermand et al. 2008), from zooplankton (Siokou-Frangou and Papathanassiou 1991; Hwang et al.2010) to fish (Wets et al., 2011). However, such dynamics both in terms of community composition and trophic relationships have never been investigated at the scale of the whole Adriatic basin. Zooplankton play a key role in marine ecosystems, forming the base of marine food web because of the diversity of their functions. Zooplankton is a link between primary producers of organic matter and the higher-order consumers, it provides grazing control on phytoplankton blooms (Kiørboe 1993) and helps regulating fish stocks (Beaugrand et al. 2003), being this last aspect of crucial importance in the Adriatic basin. Because of these important zooplankton functions, a better understanding of their distribution and the patterns of their response to changes in the chemical and physical properties of marine waters is essential, especially under a global warming scenario, being zooplankton sensitive beacon of climate change (Richardson, 2008). Moreover, trophic relationships in pelagic ecosystems are complex and complicated by the large degree of omnivory of most zooplanktonic species (Bode and Alvarez-Ossorio, 2003), which may feed on similar diets composed of a mixture of phytoplankton, detritus, and microplankton (e.g., Stoecker and Capuzzo, 1990; Irigoien et al., 1998; Batten et al., 2001). Several experimental studies allowed zooplankton (mostly copepods) to be categorised from pure carnivores to omnivores with a variety of mixtures of algae and animal prey up to strictly herbivore species (Irigoien et al., 1998; Batten et al., 2001; Halvorsen et al., 2001). Such variety in the diet makes the quantification of flows between compartments or trophic levels difficult. In the last decades, stable isotope analyses (SIA) have been widely used in food-web studies, different studies dealt with high taxonomical groups of zooplankton (Burd et al., 2002; Blachowiak-Samolyk et al., 2007; Tamelander et al., 2008), while few investigations were focused on low taxonomical resolution (Koppelmann et al., 2003; Rumolo et al., 2017), essential to disentangle the food web structure of pelagic communities (Fanelli et al., 2011). Analysis of stable isotope composition provides indications of the origin and transformations of organic matter. Stable isotopes of carbon and nitrogen integrate short-term variations in diet and thus



are less subject to temporal bias. The $\delta^{15}N$ in tissues of consumers are typically greater by 2–3‰ relative to their prey
and can be used as a proxy of the trophic level of organisms (Owens, 1987), while $\delta^{13}C$ may act as a useful indicator of
primary organic carbon sources of an animal's diet, as tissues tend to be rather weakly enriched in $^{13}C$ at progressively
higher trophic levels (1‰).
In this context the main aim of this study is to analyse mesoscale variations in the assemblage structure and
trophodynamics of mesozooplankton communities in the whole basin. Additionally, considering the complex
hydrological condition of the basin, characterised by such contrasting oceanographic settings from north to south, here
we explored and identified which environmental variables best explain the observed patterns.

## 2 Materials and Methods

### 2.1. Study area

The Adriatic Sea is an elongated semi-enclosed basin, with its major axis in the northwest–southeast direction, located in
the central Mediterranean, between the Italian peninsula and the Balkans (**Figure 1**). It is 800 km long and 150-200 km
wide. It has a total volume of 35,000 km³ that belongs for 5% to the Northern basin, 15% to the middle basin and 80% to
the Southern basin. The Northern Adriatic is very shallow, with an average depth of 35 m with a very gradual topographic
slope along its major axis and it is characterized by strong river runoff, being the Po the second main contributor (about
20%) to the whole Mediterranean river runoff (Struglia et al., 2004).


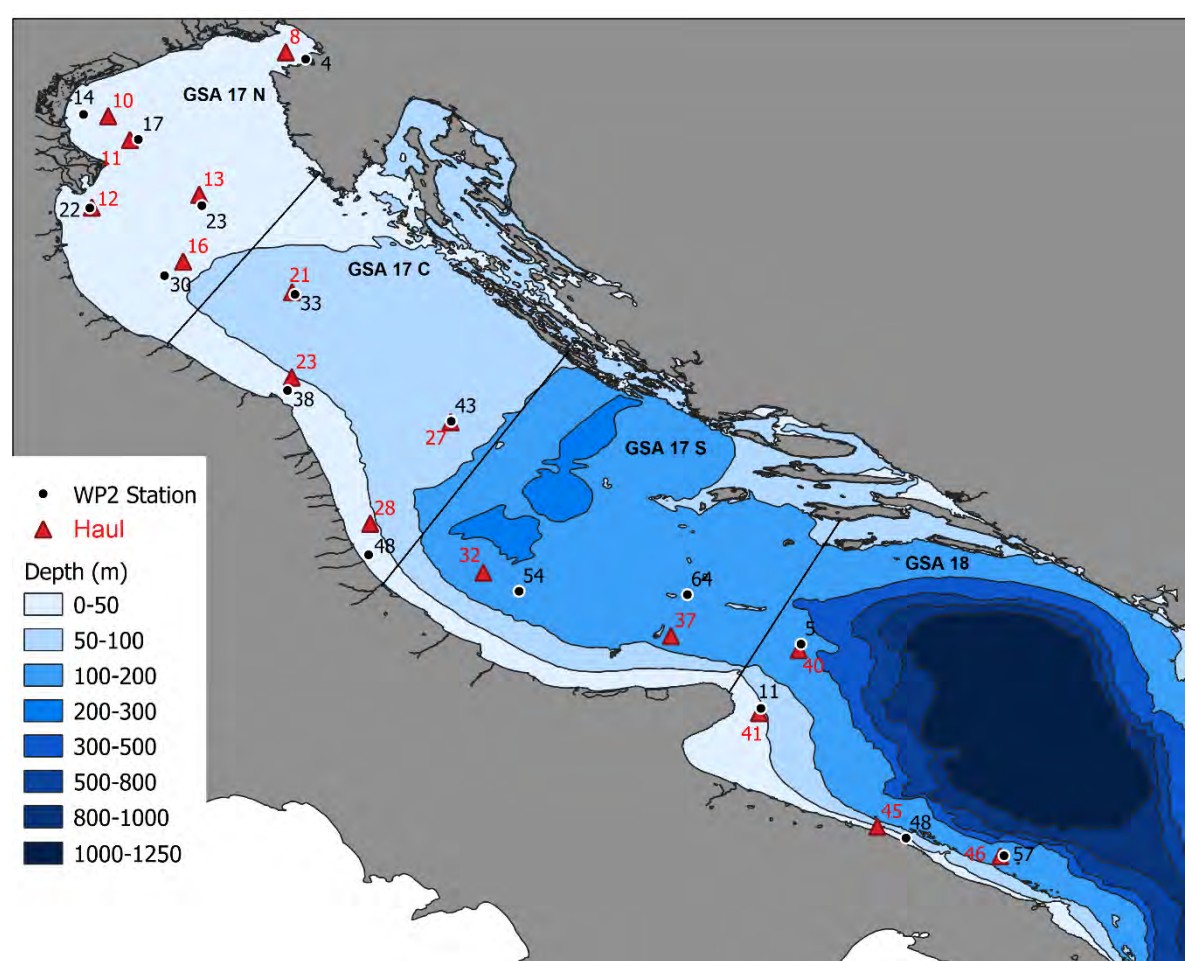

**Figure 1:** Map of the study area with indication of WP2 net stations (black dots) and mid-water trawl hauls.

Due to this input, there is a positive water balance of 90-150 km³ that is exported to the Mediterranean. The turnover time for the whole basin is 3-4 years (Artegiani *et al.*, 1997; Marini, Bombace and Iacobone, 2017). The middle Adriatic is a transition zone between northern and southern sub-basins, with the two Jabuka/Pomo depressions reaching 270 m depth. The southern sub-basin is characterized by a wide depression about 1200 m in depth. Water exchange with the Mediterranean takes place through the Otranto Strait, which has an 800 m deep sill (Artegiani *et al.*, 1997; Marini, Bombace and Iacobone, 2017). The Adriatic is a temperate warm sea, with surface temperature ranging from 6 ºC in the northern part in winter to 29 ºC, in summer. Even the temperatures of the deepest layers are, for the most part, above 10 ºC. The South Adriatic is warmer than its central and northern parts during winter. In other seasons, the horizontal temperature distribution is more uniform (Artegiani *et al.*, 1997; Marini, Bombace and Iacobone, 2017).



Water circulation in the Adriatic is mainly driven by dominant winds (Bora and Scirocco) that cause a cyclonic circulation,
with three closed circulation cells (one for each sub-basin). Three different water masses dominate the basin circulation:
the Adriatic Surface Water (AdSW), the Levantine Intermediate Water (LIW) and the Adriatic Deep Water (AdDW),
which branches out in Northern (NAdDW), Middle (MAdDW) and Southern (SAdDW) Adriatic Deep Water. The
hypersaline LIW is formed in the Levantine Basin and experiences a salinity decrease on its way to the Adriatic. The
AdDW are formed in the Adriatic basin and the NAdDW in the Northern part; due to its high density, it fills up the
Jabuka/Pomo Pit and only occasionally spreads to the Southern Adriatic. The MAdDW is formed in the Jabuka/Pomo Pit
area, when there is no intensive north-westward flow, (*i.e.* during periods of low Mediterranean water inflow). The
SAdDW originates in the South Adriatic Pit. As mentioned above, the Adriatic is a very productive basin, compared to
the rest of the Mediterranean. Despite being only the 5% of the total Mediterranean surface area, the Adriatic Sea produces
about 15% of total Mediterranean landings (and 53-54% of Italian landings), with a fish production density of 1.5 t/km²,
which is three times the Mediterranean density (Marini, Bombace and Iacobone, 2017). This impressive feature is shaped
by three main factors: river runoff, shallow depths and oceanographic structure. Rivers can indeed provide nutrients,
which favour phytoplanktonic blooms, thus causing a bottom-up effect of the whole trophic chain. Rivers can also provide
suspended particulate organic matter and organic detritus, that feed numerous particulate feeders and detritivores, such as
bivalves (which is one of the main fisheries of the North Adriatic Sea). The wide continental shelf favours a short trophic
chain that likely improve the efficiency of energy transfer from lower trophic levels to higher ones. Moreover, the structure
of the basin allows water mixing during winter, especially in North and Middle Adriatic, transferring nutrients from
sediments to the water column. From a fishery management point of view, the General Fishery Commission for the
Mediterranean (GFCM) has divided the basin in two Geographical Sub-Areas (GSAs), the GSA 17, encompassing the
northern and the middle sub-basin and the GSA 18, including the southern part.
**2.2 Zooplankton collection and analysis**
Samples for this study were collected on board R/V "G. Dallaporta" during the acoustic survey MEDIAS 2019 GSA 17
and GSA 18, that took place in June-July 2019, in the Adriatic Sea (Leonori *et al*., 2020), within the framework of the
MEDIAS (MEDiterranean International Acoustic Surveys) project. MEDIAS coordinates the acoustic surveys performed
in the Mediterranean and Black Sea to assess the biomass and spatial distribution of small pelagic fish (MEDIAS, 2019)
(http://www.medias-project.eu).
Zooplankton samples were collected through 200 μm-mesh size WP2 net, with a circular mouth of 57 cm diameter and
2.6 m long, equipped with a MF 315 flowmeter to estimate the volume of filtered water. Vertical tows were performed
with a towing speed of 1 m/s, starting from three meters above the bottom, to the surface. Sampling stations were located
along acoustic sampling transects (Figure 1).
Zooplankton samples near the fishing hauls were subsampled: half of each sample was frozen at -20 °C, to be used for
this study, while the other half was preserved in formalin. Concurrently with each vertical plankton haul, a CTD cast was



performed, to acquire information on the oceanographic parameters of the chosen site. For the purpose of this study, the
whole Western Adriatic (GSA 17 and GSA 18) has been divided in four different sub-areas, based mainly on
oceanographic characteristics: 1. GSA 17 North (GSA17N), characterized by a low depth, up to about 60 m, and mostly
influenced by the Po River input; 2. GSA 17 Central (GSA17C), with deeper bottoms, up to about 100 m; 3. GSA 17
South (GSA17S), characterized by the presence of the Pomo Pit and 4. GSA 18, characterized by the presence of the
South Adriatic Pit and the Otranto Channel. For each sub-area, hauls that were representative of variations in
oceanographic characteristics (mainly depth, distance from the coast and latitude) were selected. Such differences were
first tested by one-way PERMANOVA (Permutational Multivariate Analysis of Variance; Anderson et al., 2008) and if
no significant differences were found, sub-areas were merged for the following analyses.
Selected zooplankton samples were analysed in the laboratory to characterize the planktonic community. First, the frozen
sample was defrosted and filtered with 200 μm sieve and the obtained mass was weighted. Then samples were quickly
sorted, and larger animals isolated for first and placed in Petri dishes located on ice, in order to preserve tissue integrity.
Individuals were than identified to the lowest taxonomic level possible and stored for subsequent analysis. About 10% of
the sample was therefore weighted and all organisms in the sub-sample were identified to the lowest taxonomic level
possible.
All identified taxa were then counted and weighted with an analytical weight scale, to obtain abundance and biomass
estimations.
**2.3. Samples preparation for stable isotope analyses**
The most abundant taxa in each sample were prepared for stable isotope analyses. Selected taxa were oven-dried for 24
hours at 60 °C. Dried samples were converted to a fine powder with a mortar and pestle. For each taxon, three replicates
(when possible) were weighted (ca 0.3-1.3 mg) and placed into tin capsules. Since it was not possible to obtain enough
material of a single taxon for stable isotope analyses from stations 22_17 and 38_17, a bulk of the whole mesoplankton
community of the stations was prepared for the analyses. Acidification of samples prior to stable isotope analyses is
usually regarded as a standard procedure, since inorganic carbon could lead to an increase of $\delta^{13}C$, because it is
isotopically heavier than most carbon of organic origin and could reflect the isotopic signature of environmental carbon
(Schlacher and Connolly, 2014). However, for this study, no acidification was carried out, as this procedure generally
reduces sample biomass, leading to too little matter available for isotope analyses. Moreover, some authors revealed
negligible differences between acidified and not acidified samples (Rumolo *et al.*, 2018). However, in order to have an
indication of the possible bias, only one species was acidified, *Euchaeta sp.*, which is a very abundant copepod in Adriatic
communities. This taxon was also chosen because it has a more calcified exoskeleton and it was abundant enough to
undergo this process. Half of the sample was acidified with HCl 1M, by adding it drop by drop to the sample until bubble



cessation, then samples were oven-dried again at 60 °C for 24 h. The other half, for the analysis of $\delta^{15}N$, was not acidified,
as several studies demonstrated that the acidification procedure can alter nitrogen isotopic signature (Kolasinski, Rogers
and Frouin, 2008). Then, six replicates of each sub-samples were prepared for isotope analyses. Samples were analysed
through an elemental analyser (Thermo Flash EA 1112) for the determination of total carbon and nitrogen, and then
analysed for $\delta^{13}C$ and $\delta^{15}N$ in a continuous-flow isotope-ratio mass spectrometer (Thermo Delta Plus XP) at the
Laboratory of XX of the University of Palermo (Italy). Stable isotope ratio was expressed, in relation to international
standards (atmospheric $N_2$ and PeeDee Belemnite for $\delta^{15}N$ and $\delta^{13}C$, respectively), as:

161                    $$\delta^{13}C \text{ or } \delta^{15}N: [(R_{sample}/R_{standard})-1]*10^3$$

where $R = {}^{13}C/{}^{12}C$ or ${}^{15}N/{}^{14}N$. Analytical precision based on standard deviations of internal standards (International
Atomic Energy Agency IAEA-CH-6; IAEA-NO-3; IAEA-N-2) ranged from 0.10 to 0.19‰ for $\delta^{13}C$ and 0.02 to 0.08‰
for $\delta^{15}N$.
**2.4. Community data analyses**
Zooplankton abundance and biomass were standardized to a constant value. The adopted constant was the volume of
water filtered by the net, calculated as follows:

168                    $$V(m^3) = A \cdot B \cdot C$$

where A is the number of spins from the flowmeter, B is the number of meters travelled with each spin (given by the
manufacturer) and C is the area of the net mouth ($m^2$). When flowmeter data were not available (due to misfunctioning),
the volume was calculated as a mean value of similar nearby stations. Zooplankton abundance was expressed as number
of individuals per $m^3$, while zooplankton biomass was expressed as mg of wet weight (WW) per $m^3$.
First, the Shannon-Wiener diversity index of each station was calculated. Then, total biomass, total abundance and $H'$
diversity index were tested by univariate PERMANOVAs. Tests were run on Euclidean distance resemblance matrixes
of untransformed data and using a two-way design with sub-area as a fixed factor with three levels (GSA17N, GSA17C-
S and GSA18) and inshore-offshore location as a fixed factor with two levels (inshore *vs.* offshore), crossed within each
other, in order to assess the presence and significance of differences between stations. As the preliminary analyses showed
no significant differences between samples from GSA17C and GSA17S, these two sub-areas were merged, allowing to
use a crossed design, otherwise impossible due to the absence of the "inshore" level within the sub-area GSA17S.
Univariate PERMANOVA test were run under 9999 permutations, with permutation of residuals under a reduced model,
as permutation method, significant p-values were set at $p<0.05$.
In order to test for differences among areas and inshore *vs.* offshore communities a PERMANOVA test was performed
on the Bray-Curtis resemblance matrix of 4$^{th}$-root transformed abundance zooplankton data, using the same design



described for univariate analyses. A CAP analysis (Canonical Analysis of Principal coordinates, Anderson and Willis,
2003) was then run to visualize the observed pattern, on the factor found to be significant by PERMANOVA.
A SIMPER analysis was carried out according to the same sampling design to identify the most typifying taxon
contributing to the average similarity/dissimilarity among sub-areas and inshore *vs*. offshore locations. This was
conducted using Bray-Curtis similarity, with a cut-off for low contribution at 60%.
In order to identify the environmental drivers of zooplanktonic communities and their structure across the sampling area,
biotic data were correlated to environmental variables. Environmental data considered were pressure (db), temperature
(°C), fluorescence (µg/l), turbidity (NTU), dissolved oxygen (expressed as ml/l and saturation percentage), salinity (PSU)
and density (km/m$^3$). All data were collected through a CTD for each station. Environmental data were tested for
collinearity among variables by using a Draftsman plot, with fluorescence, Dissolved $O_2$ concentration (DO, ml/l), % of
$O_2$ saturation and turbidity data being Log (X+1)-transformed to fit a linear distribution in the Draftsman plot. Finally, a
DistLM (Distance based linear models) was run with temperature, fluorescence, turbidity, oxygen and salinity as
environmental variables, using step-wise as selection procedure and AIC (Akaike Information Criterion) as selection
criterion.

## 2.5.    Stable isotopes data analysis

Since lipids can alter the values of $\delta^{13}C$ (Post *et al.*, 2007), samples with high lipid concentration can be defatted to avoid
$^{13}C$ depletion. However, lipid extraction can alter $\delta^{15}N$ values, can complicate sample preparation and reduce samples
availability, a crucial point when analysing small animals. For these reasons, $\delta^{13}C$ of samples rich in lipids was normalized
according to Post equation (Post *et al.*, 2007):
$$\delta^{13}C_{normalized} = \delta^{13}C_{untreated} - 3.32 + 0.99\ C/N_{sample}$$
C/N ratio was used as a proxy of lipid content, because their values are strongly related in animals (Post *et al.*, 2007). In
particular, the normalization was applied to samples with a C/N ratio > 3, according to Post *et al.* (2007).
A hierarchical cluster analysis (Euclidean distance, average grouping methods) on the bivariate matrix of $\delta^{13}C$ and $\delta^{15}N$
mean values of each taxon was performed in order to elucidate the planktonic food web structure. Obtained clusters were
also compared with literature data on the trophic guild of analysed taxa. Five main trophic groups were established a
priori: primary consumers (PC), omnivores of type 1 (OMN1), encompassing mostly herbivore species, but that can feed
also small particles and ciliates, omnivores of type 2 (OMN2), similarly to OMN1 but with greater preference for small
zooplankton, carnivores (CAR) and parasite species (PAR). Differences among groups were tested by means of a one-
way PERMANOVA test with "trophic group" (with four levels) as fixed factor.
The trophic level of the different species was estimated according to Post (2002) as:
$((\delta^{15}N_i - \delta^{15}N_{PC})/TEF) + \lambda$





where $\delta^{15}N_i$ is the $\delta^{15}N$ value of the taxon considered, $\delta^{15}N_{PC}$ is the $\delta^{15}N$ values of a primary consumer, *i.e.* an herbivore
or a filter feeder, used as baseline of the food web, TEF is the trophic enrichment factor which is considered varying
between 2.54 (Vanderklift and Ponsard, 2003) and 3.4 (Vander (*e.g.* Vander Zanden and Rasmussen, 2001; Post, 2002)
and here is assumed to be 2.54 for low trophic level species, according to Fanelli et al (2009; 2011), and $\lambda$ is the trophic
position of the baseline, which is 2 in our case.
Then, differences in the isotopic composition of the overall communities by sub-area and inshore *vs.* offshore
communities were tested by two-way PERMANOVA on the same design used for assemblage analysis. The same
procedure was also used to perform univariate two-way PERMANOVA and one-way PERMANOVA with pairwise test
for the $\delta^{13}C$ and $\delta^{15}N$ values, separately.
Finally, maximum likelihood standard ellipses were created for the $\delta^{13}C$ and $\delta^{15}N$ values following Jackson et al. (2011)
to assess the community niche width in the different sub-areas. In addition to standard ellipse area (SEA; contain ca. 40%
of the data and represent the core isotopic niche) and standard ellipse areas corrected for small sample size (SEAc),
traditional convex hulls and four Layman metrics were also estimated (Layman et al., 2007). Specifically, we calculated
TA, which is the area of convex hull containing, in the case of SIBER (Stable Isotope Bayesian Ellipses in R, Jackson et
al., 2011), the means of the populations that comprise the community, d15N_range that is the distance in units between
the min and max y-axis population means, d13C_range, *i.e.* the distance in units between the min and max x-axis
population means, and CD which is the mean distance to centroid from the means. Ellipse sizes were compared between
groups (*i.e.* sub-areas) using Bayesian inference techniques.
All analyses were run using the software PRIMER7&PERMANOVA+ (Anderson *et al.,* 2008; Clarke and Gorley, 2006)
and within the jags and SIBER packages in R 4.1.0 (www.r-project.org).


**3.    Results**
**3.1. Zooplankton community and spatial changes**
A total of 52,016 specimens belonging to 113 taxa were collected through the WP2 sampling (**Table S1**). Zooplanktonic
communities in the whole area were dominated by small copepods of the genus *Acartia* (mostly *A. tonsa*), *Oncaea*,
*Oithona* (mainly *O. similis*) and copepodites. Abundant large copepods were Calanoida belonging to the genera *Euchaeta*,
*Calanus*, *Centropages* and *Temora*. Since samples were frozen on board after collection, a quite considerable number of
specimens (particularly amphipods and mysids and those taxa/specimens characterized by soft carapace) were damaged
and therefore hard to identify at species level. Generally, they were identified to order level or indicated as "damaged
unid." in **Table S1**. Other common crustaceans were hyperiids, such as *Lestrigonus schizogeneios* and *Phronima*



*atlantica*, decapod larvae (mainly zoeae and megalopae), mysids and euphausiids. Among non-crustaceans, molluscs
were quite common, both as larvae of benthic organisms and adult pteropods. Chaetognatha were also locally abundant.
Gelatinous zooplankton was represented mainly by thaliaceans and calycophorans, while ichthyoplankton was not very
abundant, with few fish eggs and larvae found.
Zooplankton abundance and biomass varied according to geographic sub-area decreasing from the Northern to the
Southern Adriatic (**Figure 2a-b**), being significant only differences in abundance, while inshore-offshore differences were
not, neither for abundance nor for biomass (**Table 1**).



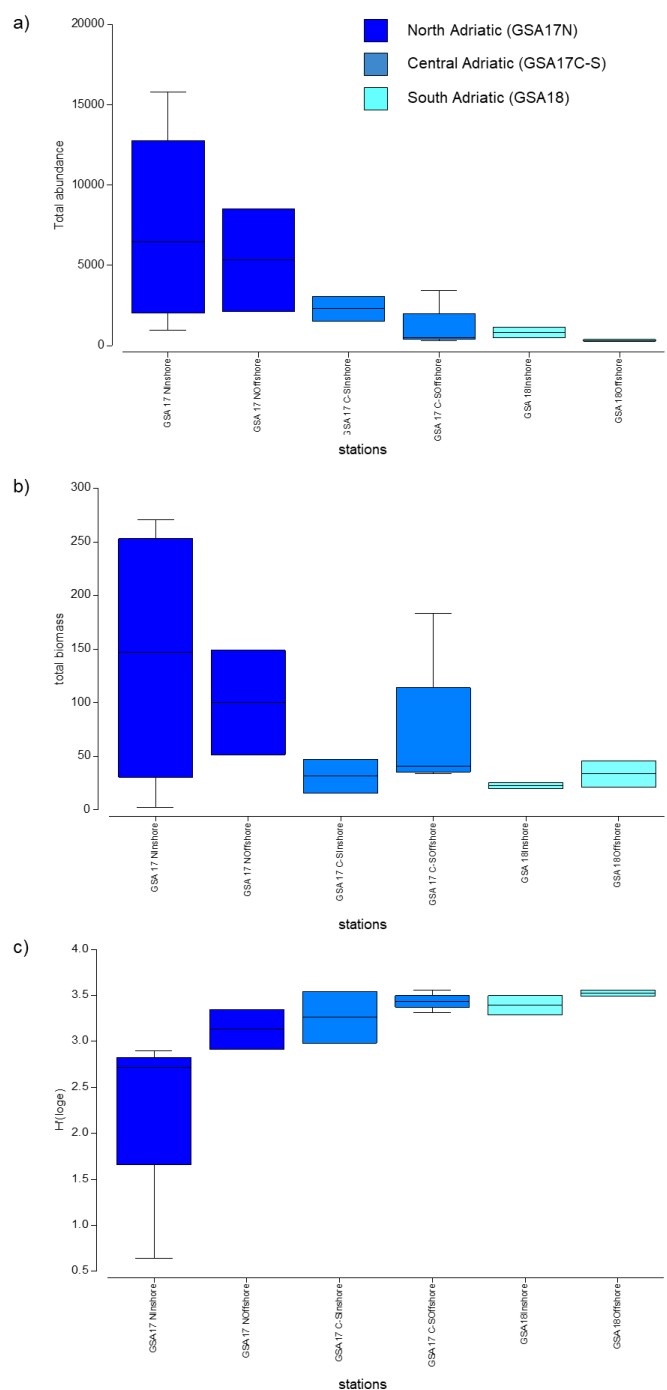

**Figure 2.** Total abundance (N ind./m³, a), total biomass (mg WW/m³, b) and diversity (*H'*, c) of mesozooplankton at each group of
stations by sub-area and distance from the coast (inshore *vs*. off-shore stations). Colours define the different sub-areas.





Diversity (in terms of *H'*) increased southward (**Figure 2c**), although differences were not significant for any of the
investigated factors. *H'* values were on average 3.25±0.31, with the only exception of station 22_17, located in the
GSA17N inshore, in front of the Po delta, showing the lowest *H'* value (0.64).

**Table 1.** PERMANOVA results of univariate analyses carried out on zooplankton abundance, biomass and diversity (in terms of *H'*
index).

| Source | df | Abundance | | | Biomass | | | Diversity (*H'*) | | |
|---|---|---|---|---|---|---|---|---|---|---|
| | | MS | Pseudo-F | P(MC) | MS | Pseudo-F | P(MC) | MS | Pseudo-F | P(MC) |
| Sub-area | 3 | 11.81 | 202.11 | 0.0001 | 0.86 | 39.42 | 0.0012 | 0.55 | 2.65 | 0.31 |
| Inshore-Offshore | 1 | 0.63 | 0.19 | 0.32 | 0.1 | 0.16 | ns | 0.46 | 1.1 | 0.45 |
| Sub-areaxInshore-Offshore | 2 | 0.02 | 0.01 | 0.86 | 0.01 | 0.02 | ns | 0.2 | 0.49 | 0.57 |
| Residuals | 9 | 3.26 | | | 0.65 | | | 0.42 | | |
| Total | 15 | | | | | | | | | |

ns=not significant difference

PERMANOVA revealed that differences in zooplanktonic communities, based on geographic sub-areas and inshore-
offshore factor were significant, while any significant differences occurred for the interaction factor (**Table 2a-b**).
**Table 2.** PERMANOVA results of multivariate analysis on zooplankton abundance, a) main test, b) pairwise comparisons for factor
"sub-area"
a)

| Source | df | MS | Pseudo-F | P |
|---|---|---|---|---|
| Sub-area | 2 | 3495 | 3.01 | 0.002 |
| Inshore-offshore | 1 | 2645.8 | 2.28 | 0.02 |
| Sub-areaxInshore-offshore | 2 | 1011.8 | 0.87 | 0.62 |
| Residuals | 10 | 1159.9 | | |
| Total | 15 | | | |

b)

| Groups | t | P |
|---|---|---|
| GSA17N *vs.* GSA17C-S | 1.55 | 0.02 |
| GSA17C-S *vs.* GSA18 | 1.40 | 0.03 |
| GSA17N *vs.* GSA18 | 2.15 | 0.01 |



The CAP plot showed a clear separation between samples from the GSA17N and all the other stations, on the first axis
(**Figure 3**).





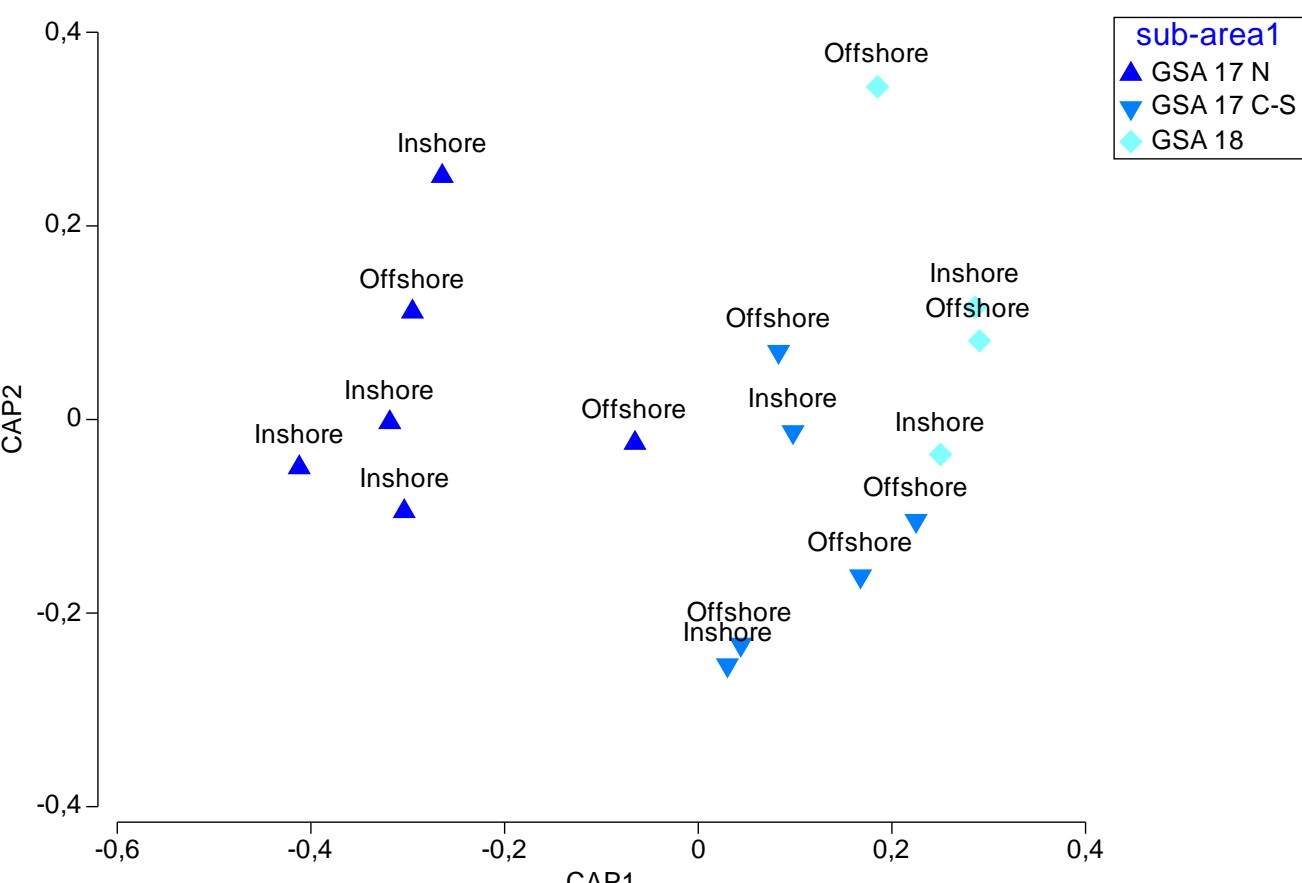

**Figure 3**. CAP plot of the mesozooplanktonic communities of the Adriatic basin by sub-area and inshore *vs.* offshore location, based on abundance data. Colours indicate the sub-areas, as described in the text.

SIMPER analysis (**Table 3a**) showed that *Acartia* sp., *Oithona* sp., unidentified Calanoida, mostly composed by copepodites, mainly contributed to dissimilarity from GSA17N *vs.* GSA17C-S. Bivalve and gastropod larvae, together with *Acartia* sp., *Oithona* sp. and unidentified Calanoida were the main responsible for the dissimilarity between the subareas GSA17C-S and GSA18. Overall, the inshore zooplanktonic communities were mostly typified by *Acartia* sp., gastropod larvae, copepodites, *Podon* sp. and *Centropages typicus*, while the offshore ones were mainly characterised by Calycophorae, *Calanus helgolandicus* and *Evadne tergestina* (**Table 3b**).





**Table 3.** Results of SIMPER analysis examining a) dissimilarity between contiguous pair of sub-area groups across all Inshore *vs.* offshore groups, and b) dissimilarity between Inshore *vs.* offshore groups across all sub-areas, with a 50% cut-off for low contribution.

a)

| GSA17N *vs.* GSA17 C-S | Average dissimilarity = 52.29 | | | | |
|---|---|---|---|---|---|
| | GSA17N | GSA17C-S | | | |
| Taxon | Av.Abund | Av.Abund | Av.Diss | Contrib% | Cum.% |
| *Acartia* spp. | 6.04 | 2.7 | 3.98 | 7.61 | 7.61 |
| *Oithona* spp. | 4.94 | 4.12 | 2.63 | 5.03 | 12.65 |
| *Penilia avirostris* | 1.48 | 1.91 | 2.33 | 4.45 | 17.1 |
| Copepoda unid. | 6.04 | 3.62 | 2.12 | 4.06 | 21.16 |
| *Evadne spinifera* | 2.43 | 0.59 | 2.09 | 4 | 25.16 |
| Gasteropoda larvae | 2.33 | 1.64 | 1.55 | 2.97 | 28.13 |
| "*Calanus*-like" copepods | 0.26 | 2.01 | 1.52 | 2.9 | 31.03 |
| *Centropages typicus* | 2.51 | 1.65 | 1.44 | 2.76 | 33.79 |
| *Euterpina acutifrons* | 1.82 | 0.52 | 1.42 | 2.71 | 36.5 |
| *Euchaeta* sp. | 0.12 | 1.84 | 1.41 | 2.71 | 39.21 |
| *Podon* spp. | 3.37 | 2.17 | 1.29 | 2.47 | 41.68 |
| *Evadne tergestina* | 0.59 | 0.31 | 1.27 | 2.42 | 44.1 |
| *Temora longicornis* | 1.12 | 0.35 | 1.17 | 2.24 | 46.34 |
| *Oncaea* sp. | 1.43 | 2.5 | 1.09 | 2.08 | 48.42 |
| *Gaetanus tenuispinus* | 0.1 | 0.55 | 0.92 | 1.76 | 50.18 |
| GSA17C-S *vs.* GSA18 | Average dissimilarity = 49.15 | | | | |
| | GSA17 C-S | GSA 18 | | | |
| Bivalvia larvae | 2.37 | 0.47 | 2.16 | 4.4 | 4.4 |
| *Oithona* sp. | 4.12 | 2.04 | 2 | 4.07 | 8.47 |
| *Acartia* sp. | 2.7 | 0.71 | 1.97 | 4 | 12.47 |
| Copepoda unid. | 3.62 | 4.17 | 1.55 | 3.16 | 15.63 |
| Gasteropoda larvae | 1.64 | 0.53 | 1.41 | 2.87 | 18.5 |
| *Penilia avirostris* | 1.91 | 1.34 | 1.33 | 2.71 | 21.21 |
| "*Calanus*-like" copepods | 2.01 | 0.97 | 1.23 | 2.5 | 23.71 |
| *Podon* sp. | 2.17 | 1.49 | 1.22 | 2.48 | 26.19 |
| Appendicularia | 1.04 | 1.2 | 1.19 | 2.41 | 28.6 |
| *Temora stylifera* | 0.81 | 0.93 | 1.06 | 2.17 | 30.77 |
| *Coryceus* sp. | 1.1 | 1.55 | 1.05 | 2.14 | 32.91 |
| Chaetognatha | 1.22 | 2.13 | 1 | 2.03 | 34.94 |
| Thaliacea | 0.88 | 1.05 | 0.97 | 1.98 | 36.92 |
| *Creseis acicula* | 0.93 | 1.27 | 0.95 | 1.93 | 38.84 |
| *Engraulis encrasicolus* eggs | 0.86 | 0.18 | 0.85 | 1.73 | 40.57 |
| *Oncaea* sp. | 2.5 | 2.45 | 0.81 | 1.66 | 42.23 |
| *Centropages typicus* | 1.65 | 1.1 | 0.81 | 1.66 | 43.88 |





| Taxon | | | | | |
|---|---|---|---|---|---|
| *Lucicutia flavicornis* | 0.12 | 0.62 | 0.78 | 1.58 | 45.46 |
| Calycophorae | 0.62 | 0.89 | 0.75 | 1.52 | 46.98 |
| *Microsetella* sp. | 0.41 | 0.63 | 0.69 | 1.41 | 48.39 |
| *Calanus helgolandicus* | 1.52 | 1.12 | 0.66 | 1.34 | 49.73 |
| *Calanus minor* | 0.56 | 0.94 | 0.66 | 1.34 | 51.06 |


b)

| Groups Inshore & Offshore | Average dissimilarity = 49.12 | | | | |
|---|---|---|---|---|---|
| | Inshore | Offshore | | | |
| Taxon | Av.Abund | Av.Abund | Av.Diss | Contrib% | Cum.% |
| *Acartia* spp. | 4.1 | 2.81 | 2.33 | 4.73 | 4.73 |
| *Penilia avirostris* | 1.89 | 1.32 | 1.96 | 3.98 | 8.71 |
| *Oithona* spp. | 3.92 | 3.89 | 1.94 | 3.95 | 12.67 |
| Gasteropoda larvae | 2.31 | 0.93 | 1.59 | 3.23 | 15.9 |
| Copepoda unid. | 5.43 | 3.89 | 1.57 | 3.19 | 19.09 |
| *Podon* spp. | 3.22 | 1.68 | 1.5 | 3.06 | 22.16 |
| *Centropages typicus* | 2.31 | 1.37 | 1.41 | 2.88 | 25.04 |
| Calycophorae | 0.23 | 1.17 | 1.33 | 2.71 | 27.75 |
| *Evadne tergestina* | 0.36 | 0.44 | 1.18 | 2.4 | 30.15 |
| *Calanus helgolandicus* | 0.52 | 1.71 | 1.14 | 2.32 | 32.47 |
| *Oncaea sp.* | 1.77 | 2.41 | 1.11 | 2.26 | 34.73 |
| *Evadne spinifera* | 1.35 | 0.92 | 1.1 | 2.25 | 36.98 |
| Chaetognatha | 0.65 | 1.66 | 1.05 | 2.14 | 39.12 |
| "Calanus-like" copepods | 0.59 | 1.6 | 0.93 | 1.89 | 41.01 |
| Bivalvia larvae | 1.75 | 1.79 | 0.87 | 1.78 | 42.79 |
| Appendicularia | 0.35 | 1.03 | 0.85 | 1.73 | 44.51 |
| Thaliacea | 0.33 | 0.96 | 0.83 | 1.69 | 46.2 |
| Zoea Brachyura | 1.13 | 1.07 | 0.82 | 1.68 | 47.88 |
| *Temora stylifera* | 0.53 | 0.65 | 0.82 | 1.66 | 49.54 |
| *Engraulis encrasicolus* eggs | 0.58 | 0.89 | 0.81 | 1.65 | 51.19 |


**3.2.  Correlation between zooplankton data and environmental variables**
Draftsman plot allowed to assess collinearity between pair of variables at $\rho>0.7$. DO concentration (ml/l) and % of oxygen
saturation covaried, as well as density and pressure, therefore, only temperature, fluorescence, turbidity, DO and salinity
were used for DistLM analysis.
DistLM results showed that 26.9% of the variance was explained by salinity, 11% by fluorescence and 8.6% by DO, those
three variables cumulatively accounting for 46.5% of variance (**Table 4**).






**Table 4.** Results of sequential test for DistLM model

| Variable | AIC | SS(trace) | Pseudo-F | P | Prop. | Cumul. | res.df |
|----------|-----|-----------|----------|-----|-------|--------|--------|
| Salinity | 116.47 | 6643.1 | 5.15 | 0.001 | 0.27 | 0.27 | 14 |
| Fluorescence | 115.86 | 2716.6 | 2.30 | 0.032 | 0.11 | 0.38 | 13 |
| DO | 115.47 | 2126.6 | 1.93 | 0.044 | 0.09 | 0.46 | 12 |


### 3.3. Stable isotope composition of zooplankton

Stable isotope analyses provided $\delta^{13}C$ and $\delta^{15}N$ values of 26 different taxa (**Table 5**). Acidification of crustaceans was
proved to be unnecessary, as the tested samples of *Euchaeta* sp. showed little and not significant differences in $\delta^{13}C$ value
(-21.39±0.06 for untreated samples *vs*. -21.02±0.15 for acidified samples, paired T-test, T= -0.34, P=0.74).





**Table 5.** Mean values of zooplankton samples, trophic group (TG), Trophic position (TP), sub-area and number of samples analysed (N), PC = primary consumers, OMN1 = omnivores of type 1, OMN2 = omnivores of type 2, CAR = carnivores, question marks (?) indicate taxa with unknown trophic group

| Group | Code | Taxon | $\delta^{15}N\pm SD$ | $\delta^{13}C\pm SD$ | TG | TP | GSA17N | GSA17C-S | GSA18 | N |
|---|---|---|---|---|---|---|---|---|---|---|
| COPEPODA | Gte | Gaetanus tenuispinus | 2.68 | -20.44 | Unk | 2.0 | * | | | 1 |
| COPEPODA | Nmi | Nannocalanus minor | 3.53±0.37 | -20.77±0.21 | Unk | 2.3 | | * | * | 3 |
| HYPERIIDEA | Pat | Phronima atlantica | 3.58 | -20.98 | OMN1 | 2.4 | | | * | 1 |
| COPEPODA | Pab | Pleuromamma abdominalis | 3.59 | -21.14 | OMN1 | 2.4 | | * | * | 1 |
| COPEPODA | Cli | Calanus-like | 3.80±0.22 | -21.28±0.48 | Unk | 2.4 | | * | * | 2 |
| THALIACEA | Tha | Thaliacea | 3.82±0.82 | -19.56±1.16 | PC | 2.4 | | * | * | 10 |
| DECAPODA | Bra | Brachyura (Zoea) | 3.89±0.06 | -19.17±0.07 | PC | 2.5 | | * | * | 2 |
| DECAPODA | Thl | Thalassinidea (Zoea) | 4.14 | -19.92 | Unk | 2.6 | | | * | 1 |
| COPEPODA | Cai | Calanoida (copepodites) | 4.16±1.09 | -20.53±0.34 | PC | 2.6 | | * | * | 3 |
| EUPHAUSIACEA | Meg | Meganyctiphanes norvegica | 4.48±0.54 | -21.18±0.57 | OMN1 | 2.7 | | * | * | 4 |
| EUPHAUSIACEA | Eup | Euphausiacea | 4.69 | -20.39 | OMN2 | 2.8 | | | * | 1 |
| HYPERIIDEA | Lpu | Lycea pulex | 4.69 | -19.64 | CAR | 2.8 | | | * | 1 |
| COPEPODA | Par | Pareucalanus attenuatus | 4.92 | -20.01 | OMN1 | 2.9 | | | * | 1 |
| OSTEYCHTHYES | Fis | Fish larvae | 5.09±0.53 | -20.57±0.26 | CAR | 2.9 | | * | * | 4 |
| COPEPODA | Che | Calanus helgolandicus | 5.16±0.94 | -20.83±0.40 | OMN1 | 3.0 | * | * | * | 15 |
| COPEPODA | Euc | Euchaea sp. | 5.19±0.40 | -20.91±0.23 | CAR | 3.0 | | * | * | 16 |
| HYPERIIDEA | Pse | Phronima sedentaria | 5.42 | -19.6 | OMN1 | 3.1 | | | * | 1 |
| COPEPODA | Cty | Centropages typicus | 5.42±1.43 | -21.38±0.34 | OMN2 | 3.1 | * | * | * | 4 |
| COPEPODA | Tst | Temora stylifera | 5.47±1.08 | -20.17±0.56 | OMN1 | 3.1 | | * | * | 2 |
| CHAETOGNATHA | Cha | Chaetognatha | 6.19±1.40 | -19.82±0.43 | CAR | 3.4 | * | * | * | 20 |
| SIPHONOPHORA | Cal | Calycophorae | 6.21±1.25 | -20.05±1.28 | CAR | 3.4 | * | * | * | 9 |
| DECAPODA | Pen | Penaeidae (Zoea) | 6.34±1.12 | -20.54±0.41 | Unk | 3.4 | * | * | * | 4 |
| DECAPODA | Dec | Decapoda (Zoea) | 6.59±1.59 | -19.81±0.24 | Unk | 3.5 | * | * | * | 5 |
| COPEPODA | Aca | Acartia sp. | 7.86±0.86 | -21.58±0.61 | OMN2 | 4.0 | * | * | * | 3 |
| HYPERIIDEA | Lsc | Lestrigonus schizogeneios | 8.06±3.30 | -20.16±0.65 | CAR | 4.1 | | * | * | 2 |
| MYSIDA | Lgr | Leptomysis gracilis | 8.14 | -20.03 | OMN1 | 4.1 | | | * | 1 |



Cluster analysis allowed to group animals according to their $\delta^{13}C$ and $\delta^{15}N$ values, and partially with the trophic groups
previously established, based on literature data when available (**Figure 4**). One-way PERMANOVA test run on factor
"trophic groups-TG" proved significant differences (*pseudo-$F_{5,28}$*=2.57, p=0.02), with primary consumers being
significantly different from omnivores of type 2 and carnivores (PC *vs*. OMN2: t=2.67, p=0.03; PC *vs*. CAR: t=3.01,
p=0.006) and omnivores of type 1 from carnivores (t=1.98, p=0.049).

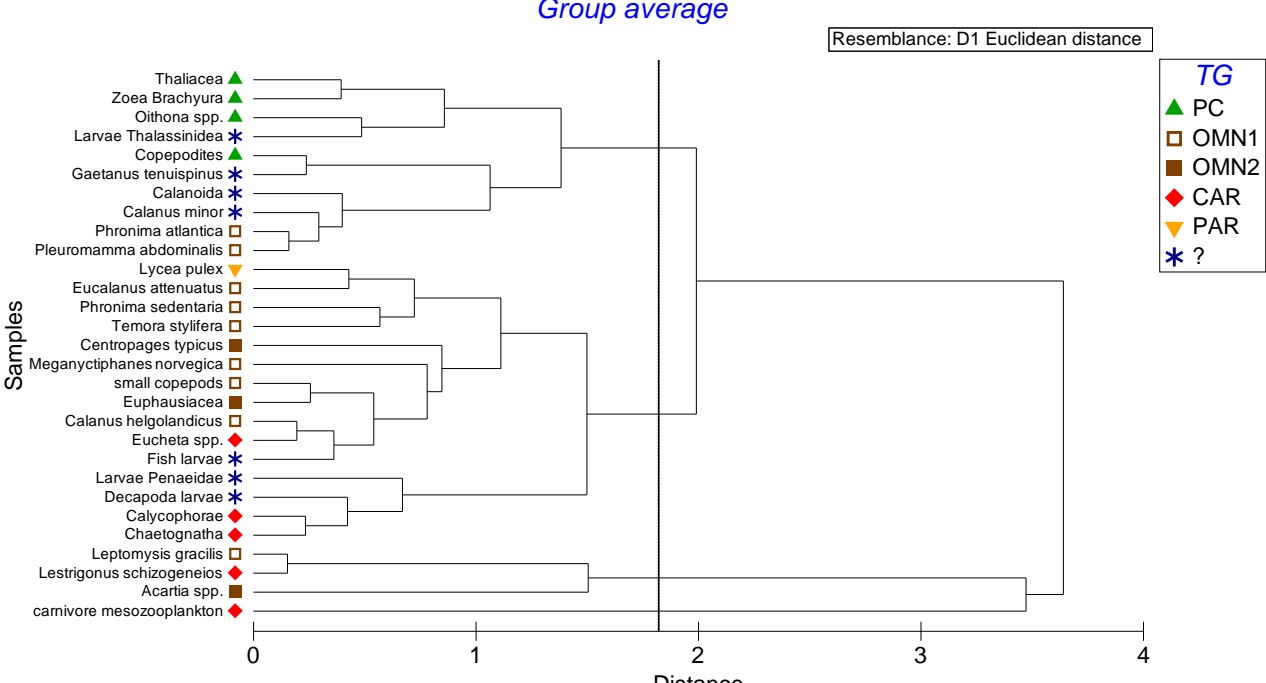


**Figure 4**. Cluster analysis on the bivariate matrix of $\delta^{13}C$ and $\delta^{15}N$ values of dominant zooplankton taxa. Colours indicate trophic
groups: OMN1 are omnivores of type 1, OMN2 are omnivores of type 2, PC are primary consumers, CAR are carnivores, question
marks (?) indicate taxa with unknown trophic group. The dashed line is placed at 1.8 distance.

The estimates of Trophic Levels (TLs), considering the $\delta^{15}N$ value of *Gaetanus spinosus* as baseline, allowed to assign
zooplanktonic taxa to 3 TLs from herbivores located at TL 2 to carnivores at TL 4 (**Table 5**).
Overall, the $\delta^{15}N$ of the mesozooplanktonic community was greater in the GSA17N, both for inshore and offshore
communities (**Figure 5**). Similarly, the median $\delta^{13}C$ value was similar among the different sub-areas, although in the
GSA17N both the greater and lower values were found in the GSA17N than in the other sub-areas, although, than for the
other communities (**Figure 5**).



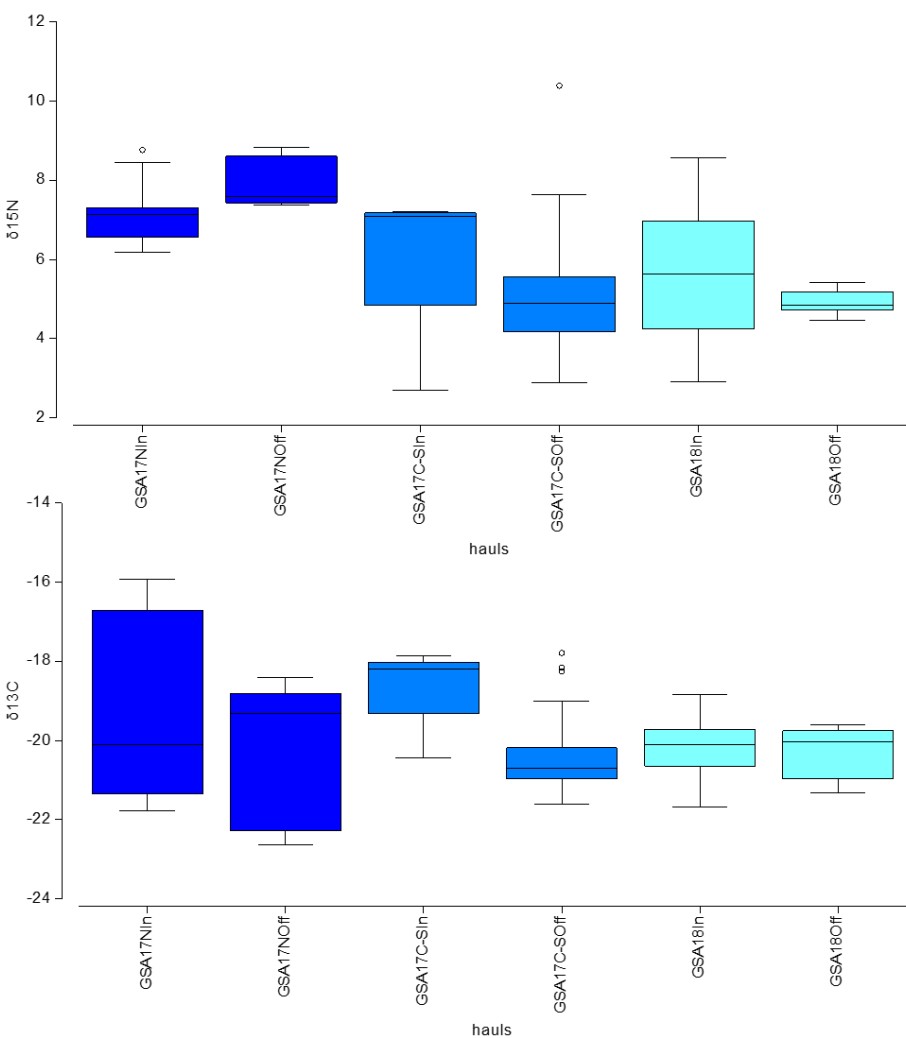


**Figure 5**. Box plot of mean $\delta^{15}N$ and $\delta^{13}C$ values of zooplanktonic taxa for each sub-area at inshore *vs.* offshore locations.

Two-way PERMANOVA on the multivariate matrix of $\delta^{13}C$ and $\delta^{15}N$ showed a significant separation according to all
factors (**Table 6a**). The pairwise comparison on sub-area factor, considering only comparisons between contiguous sub-
areas showed a significant separation between the isotopic composition of zooplanktonic taxa from the GSA17N *vs.*
GSA17C-S, but not between the GSA17C-S and GSA18. The pairwise test run on the interaction factor for pairs of level
of factor "inshore *vs.* offshore" provided evidence for significant variations in the isotopic composition between inshore
and offshore communities only for the GSA17C-S. One-way PERMANOVA tests run separately on $\delta^{15}N$ values showed
significant variation only for factor sub-area (**Table 6b**) with the isotopic composition of zooplankton from GSA17N
significantly different from that of GSA18. Conversely, $\delta^{13}C$ values significantly varied between inshore and offshore



communities and for the interaction factor (**Table 6c**), being differences between inshore and offshore communities significant in GSA17C-S.








**Table 6.** Results of PERMANOVA a) main test and pairwise comparisons for b) sub-area factor and c) interaction term for pairs of levels of factor "inshore vs. offshore" run on the Euclidean resemblance matrix of untransformed $\delta^{15}N$ and $\delta^{13}C$ values and for $\delta^{15}N$ and $\delta^{13}C$, separately.

### $\delta^{15}N\text{-}\delta^{13}C$

a)

| Source | df | MS | Pseudo-F | P |
|---|---|---|---|---|
| Sub-area | 2 | 27.52 | 10.86 | 0.0001 |
| Inshore vs. offshore | 1 | 11.10 | 4.38 | 0.01 |
| Sub-areax Inshore vs. offshore | 2 | 7.20 | 2.84 | 0.03 |
| Residuals | 121 | 2.53 | | |
| Total | 126 | | | |

b)

| Groups | t | P(perm) |
|---|---|---|
| GSA17N, GSA17C-S | 3.58 | 0.0001 |
| GSA17C-S, GSA18 | 1.41 | 0.14 |

c)

Within level 'GSA17N of factor 'subarea'

| Groups | t | P(perm) |
|---|---|---|
| Inshore vs. offshore | 0.88 | 0.40 |

Within level 'GSA17C-S' of factor 'subarea'

| Groups | t | P(perm) |
|---|---|---|
| Inshore vs. offshore | 2.95 | 0.004 |

Within level 'GSA18' of factor 'subarea'

| | t | P(perm) |
|---|---|---|
| Inshore vs. offshore | 1.10 | 0.28 |

### $\delta^{15}N$

a)

| | MS | Pseudo-F | P(perm) |
|---|---|---|---|
| | 25.33 | 16.74 | 0.0001 |
| | 1.39 | 0.92 | 0.33 |
| | 3.77 | 2.49 | 0.09 |
| | 1.51 | | |

b)

| Groups | t | P(perm) |
|---|---|---|
| GSA17N, GSA17C-S | 5.15 | 0.0001 |
| GSA17C-S, GSA18 | 0.50 | 0.62 |

c)

Within level 'GSA17N of factor 'subarea'

| Groups | t | P(perm) |
|---|---|---|
| In. Off | 0.58 | 0.57 |

Within level 'GSA17C-S' of factor 'subarea'

| Groups | t | P(perm) |
|---|---|---|
| In. Off | 4.43 | 0.0008 |

Within level 'GSA18' of factor 'subarea'

| | t | P(perm) |
|---|---|---|
| In. Off | 0.50 | 0.62 |

### $\delta^{13}C$

a)

| | MS | Pseudo-F | P(perm) |
|---|---|---|---|
| | 2.18 | 2.14 | 0.12 |
| | 9.72 | 9.52 | 0.003 |
| | 3.43 | 3.36 | 0.04 |
| | 1.02 | | |

b)

c)

Within level 'GSA17N of factor 'subarea'

| Groups | t | P(perm) |
|---|---|---|
| In. Off | 0.58 | 0.57 |

Within level 'GSA17C-S' of factor 'subarea'

| Groups | t | P(perm) |
|---|---|---|
| In. Off | 4.43 | 0.0008 |

Within level 'GSA18' of factor 'subarea'

| | t | P(perm) |
|---|---|---|
| In. Off | 0.50 | 0.62 |





Finally, the SIBER method for calculating ellipse-based metrics of niche width provided evidence of larger niche width for
the zooplanktonic community from GSA17N than GSA17C-S and GSA18 (**Figure 6** and **Table 7**). Estimated overlap by
Bayesian inference evidenced almost null overlap between GSA17N and GSA17C-S ($<10^{-15}$), while a high overlap existed
between GSA17C-S and GSA18 (0.26). The greater d15N_range was observed for GSA17C-S and GSA18 communities, while
the higher d13C_range occurred in GSA17N communities, where also CD value was the greatest (**Table 7**).

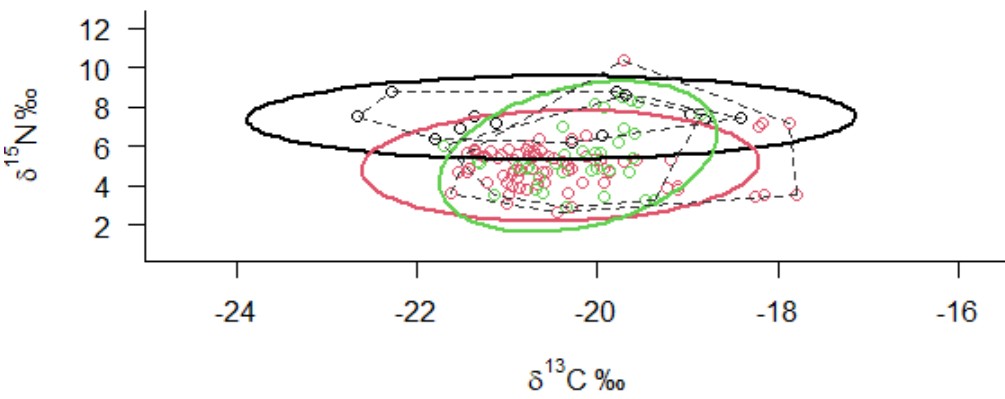

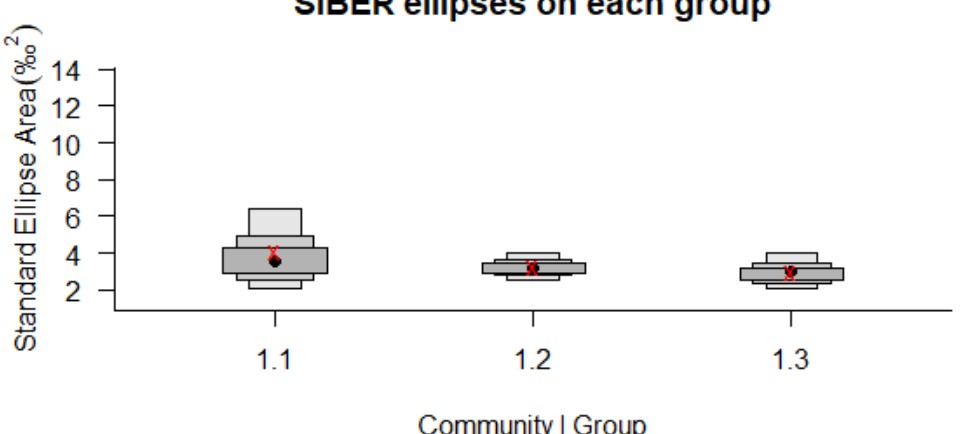

**Figure 6.** Top: Standard Ellipse Areas for the three zooplanktonic communities analysed; the black circle and symbols indicate the
GSA17N community, the red ones the GSA17C-S and the green ones the GSA18. Bottom: Credible intervals for the estimated SEAc of
the three communities, 1.1= GSA17N, 1.2= GSA17C-S, 1.3= GSA18.






**Table 7.** Estimates of Convex hulls (TA). Standard Ellipse Areas (SEA and SEAc. as corrected for low sample size). and Layman metrics
d15N_range. d13C_range and Mean Distance to Centroid (CD). calculated fotr the three sub-areas zooplanktonic communities.

| Metrics | GSA17N | GSA17C-S | GSA18 |
|---|---|---|---|
| TA | 7.96 | 19.39 | 10.38 |
| SEA | 3.74 | 3.24 | 2.92 |
| SEAc | 4.08 | 3.28 | 3.00 |
| d15N_range | 2.65 | 7.71 | 5.65 |
| d13C_range | 4.23 | 3.83 | 2.84 |
| CD | 1.93 | 1.16 | 1.42 |


## 4. Discussion
These are the first results on mesozooplankton community composition and food web structure conducted at basin scale for
the Adriatic Sea. Considering that the Adriatic Sea is one of the largest areas of occurrence of demersal and small pelagic
shared stocks in the Mediterranean (FAO, 2020), this study may represent an important piece to reconstruct the whole pelagic
food web and changes at mesoscale level across the basin. Still, considering the increasing fishing pressure in the basin together
with evidence of primary production (climate-change related) decrease after the 1980s (Solidoro et al., 2009; Mozetic et al.,
2010), this study may represent a valid baseline for future comparison on the synergic and cumulative effect of climate change
and overfishing in one of the most impacted regions within the Mediterranean Sea (Coll et al., 2012; Micheli et al., 2013).
### 4.1. Mesoscale variations in zooplankton biomass, abundance and composition
Overall, 113 taxa and 57 species have been identified during June-July 2019 in the Adriatic basin (**Table S1**). These values
were only slightly lower than those observed for the Central Adriatic at 0-50 m depths where 150 taxa were counted (Hure et
al., 2018). Such differences maybe only apparent and attributable to the storage method we used, as samples were kept frozen
for subsequent stable isotope analyses, determining a damage in many organisms, which were impossible to identify to species
or even genus level (Fanelli et al., 2011). In terms of species abundance, the most representative species were *Acartia tonsa*,
*Oithona similis* and *Centropages typicus* among copepods, and the cladocerans *Podon intermedius*, *P. polyphemoides*, *Penilia*
*avirostris*, *Evadne tergestina* and *E. spinifera*, in agreement with previous studies on the mezooplanktonic communities of the
Adriatic basin (Fonda-Umani et al., 2005; Bernardi Aubry et al., 2012).
Zooplankton biomass and abundance were higher in the Northern Adriatic Sea and slowly decreased moving towards the
Southern Adriatic. This trend was also observed by Fonda Umani (1996) and can be explained by the influence of Po River,
which can determine a high nutrient input in the Northern Adriatic favouring primary production and therefore zooplankton
growth. Notwithstanding the general primary production reduction observed in the last years (Mozetič et al., 2010) in the North
Adriatic Sea, the area is still characterised by higher phytoplankton biomass with respect to the central and the southern basin,
because of the nutrients input from the Po River. Chlorophyll-a concentration values from satellite data (**Figure 7**,
https://giovanni.gsfc.nasa.gov/giovanni) analysed from four months before the sampling period to survey simultaneous period





(July 2019), revealed indeed a peak in primary production in May 2019, two months before the sampling period, in the area in
front of the Po River delta, fuelling in turn zooplankton production (Bernardi Aubry et al., 2012).

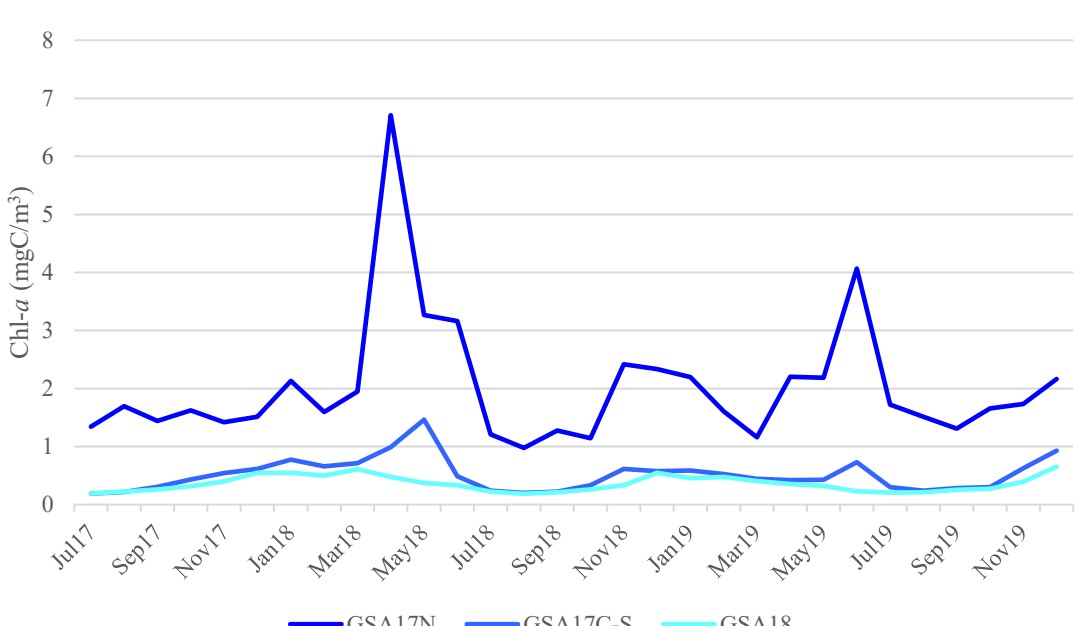



**Figure 7.** Monthly time-series area-averaged map of satellite-derived (Sensor MODIS Aqua from https://giovanni.gsfc.nasa.gov/giovanni)
Chlorophyll a concentration (mgC/m$^3$) from July 2017 to December 2019 for the three sub-areas considered in this study.

Although, in the north-western Adriatic offshore waters are less productive than inshore coastal waters and productivity of the
inshore zone decreases southward away from the Po Rivers' nutrient influx (Vollenweider et al., 1998), here we did not find
significant differences in terms of abundance and biomass between inshore and offshore communities or for the interaction
factors. Such differences were instead observed when we compared zooplanktonic communities' composition. Indeed,
multivariate analyses evidenced a clear separation of samples as function of sub-area and inshore *vs*. offshore locations, and
especially between the mesozooplanktonic community of the Northern Adriatic from the other two. This was not surprising as
the northern Adriatic is characterised by shallower and colder waters than the rest of the basin and under the influence of
riverine input, thus hosting a typical neritic community with coastal and estuarine elements. This area was dominated also by
*Acartia clausi*, *Oithona similis*, cladocerans (mostly *Evadne spinifera*), copepodites (here comprised within the "Copepoda
unid." group), gastropod larvae with some differences with respect to previous studies (Bernardi Aubry et al., 2012), in terms
of temporal shift of species maximum abundance. This could be related to the peak in primary production occurring in May
2019, quite delayed with respect to the usual pattern of the area (Kamburska and Fonda-Umani, 2009) (see **Figure 7**).



Conversely, the southern Adriatic basin, except for the Gargano promontory, being characterised by a narrow continental shelf
and a steep slope, reaching high depths close to the coasts, was dominated by typical offshore species such as tunicates,
chaetognaths, siphonophores and *Euchaeta* spp. These results were supported by Fonda Umani (1996), that identified a clear
distinction in zooplanktonic communities collected in offshore location of Northern and Central-Southern Adriatic: the
Northern Adriatic was characterized by neritic communities, with moderate biomass, while the Central and the Southern
Adriatic Sea were characterized by an "oceanic" community, with a higher abundance of carnivorous zooplankton, such as
*Euchaeta* sp., a more oceanic carnivorous genus (Razouls et al., 2021), and Chaetognatha, a phylum of carnivorous animals
quite abundant in open waters (Terazaki, 2000). Consistently, diversity was the greatest in the southern basin, with 80 taxa
(out of 113) identified, likely due to the occurrence of both neritic and oceanic species in this area and comparable to other
studies (Miloslavic et al., 2012) which included also deep stations.

### 4.2. Environmental drivers of zooplankton communities variability

Separation among samples according to sub-areas and inshore and offshore locations were consistent with the main drivers
resulted by the distance-based multivariate model, *i.e.* salinity, fluorescence and DO concentration, with salinity itself
explaining 27% of the variance. Salinity values were on average 36 psu in the Northern basin with the lowest value of 34.7
psu recorded at station 22_17 in front of the Po; salinity increased southward reaching a mean value of 38.7 psu in the southern
basin. Fluorescence values decreased southward from 2.45 µg/l to 0.77 µg/l, with the highest (4.9 µg/l) and the lowest (0.59
µg/l) values recorded at station 22_17 (in front of the Po river delta) and at station 44_18 (in the Otranto channel), respectively.
These two variables were mainly linked to freshwater inputs from the Po River and were responsible of the main separation
between the Northern Adriatic, more coastal-estuarine zooplanktonic communities, from the central and southern Adriatic,
more oceanic zooplanktonic communities. On the other hand, changes in DO which decreased southward from a mean value
of 5.32 ml/l recorded in GSA17N stations to 4.36 ml/l were observed in GSA18 CTD casts. This is in full agreement also with
the decreasing trend in zooplankton biomass from the GSA17N to GSA18. Several studies indicated that oxygen concentration
could be a limiting factor for zooplankton growth and survival (Olson, 1987; Moon et al., 2006), with inhibition of egg hatching
in some copepod species (Roman et al., 1993). DO was found to be also the driving factor of zooplanktonic communities in
the strait of Sicily (Rumolo et al., 2016)

### 4.3. Food web structure of zooplankton communities

The trophic groups highlighted by cluster analysis fully agreed with putative trophic groups established *a priori* based on
literature information and allowed to assign species with unknown feeding ecology to a trophic group. Two main groups were
evident, a first one grouping primary consumers (*i.e.* herbivore/filter feeder taxa) and copepodites assigned to omnivore of
level 1, *i.e.* taxa that may act both as primary consumer eating phytoplankton or detritus particles or shifting to small prey, *i.e.*
microzooplankton. Taxa with unknown feeding mode such as *Gaetanus tenuispinus* and small calanoids (including



copepodites) were sorted together with *Nannocalanus minor*, *Phronima atlantica* and *Pleuromamma abdominalis*, because of
their lowest δ$^{15}$N value, similar to that of filter feeders (Rumolo *et al.*, 2016). According to literature, *Phronima atlantica*
should be a carnivorous species, feeding on salp tissue (Madin and Harbison, 1977). However, Elder and Seibel (2015) also
reported feeding on host mucus, which could lower their trophic position, being more similar to the basal source, *i.e.* the
particulate organic matter or POM (Fanelli et al., 2011). Zoeae of Thalassinidea and Brachyura were also placed in this group,
close to thaliaceans, that are herbivorous filter feeders (Madin, 1974).
The second group encompass different taxa mostly carnivores and omnivores of both level 1 and 2, *i.e.* taxa that mostly prefer
animal prey but that can shift to phytodetritus when prey was scarce or competition was high (Fanelli et al., 2011). This is the
case of *Meganichtyphanes norvegica* which can vary its diet regionally and with growth, showing a preference for
phytoplankton in certain areas, seasons or when juveniles (Schmidt, 2010; Fanelli et al., 2011), or preying exclusively on
calanoids when adults or depending on energy requirements (McClatchie, 1985). Other examples of this kind are represented
by the calanoid *Calanus helgolandicus* or *Centropages typicus*. *C. typicus* is an omnivorous copepod that feeds on a wide
spectrum of prey, from small algae (3–4 μm equivalent spherical diameter) to yolk-sac fish larvae (3.2–3.6 mm length). It uses
both suspensivorous and ambush feeding strategies, depending on the characteristics of the prey (Calbet et al., 2007) Although
*C. helgolandicus* was described as an herbivore species (Paffenhoffer, 1976) some authors described density-dependent
mortality through cannibalism in *Calanus* spp., as a form of population self-limitation (Ohman and Hirche, 2001), thus pointed
out to an omnivorous feeding behaviour. Omnivorous copepods can display increased predatory behaviour in the absence of
other food (Daan, 1988), and may actively target eggs even when phytoplankton is not limiting (Bonnet et al. 2004). Finally,
a mixed group formed by the mysid *Leptomysis gracilis*, the copepod *Acartia tonsa* and the hyperiid *Lestrigonus schizogeneios,*
clustered close to other carnivore species. Hyperiids generally use gelatinous substrate for reproduction and feeding, some of
them living in symbiosis (Gasca and Haddock, 2004) other being parasite such as the genus *Hyperia* (now *Lestrigonus*). *A.*
*tonsa* may display both predatory and suspension feeding behaviour (Saiz and Kiorboe, 1995), similarly to *L. gracilis* (Fanelli
et al., 2009) and accordingly to their isotopic composition and position in the zooplanktonic food web.
The average enrichment between the different plankton taxa was greater than the mean value of 2.56 expected between adjacent
trophic levels (e.g., Vanderderklift and Ponsard, 2003; Fanelli et al., 2011) pointing to the organization of mesozooplanktonic
taxa in three trophic levels, from the copepod *Gaetanus tenuispinus*, positioned at the trophic level 2, to the highest-level
species represented by *Lestrigonus schizogeneios* and *Leptomysis gracilis*, located at the trophic level 4. Such results confirmed
other findings (Fanelli et al., 2009, 2011) about the complexity of pelagic food webs and of their lower trophic levels, calling
attention on the appropriate compartmentation of zooplankton in ecosystem modelling with the final scope of small pelagic
stock management (D'Alelio et al., 2016).

**4.4.       Spatial variability in the isotopic composition of mesozooplankton from the Adriatic basin**



Overall, stable isotope values of zooplankton differed significantly for all factors considered, with $\delta^{15}N$ values decreasing
southward, and $\delta^{13}C$ showing more constant patterns with the exception of the GSA17N. The presence of differences in isotopic
signature of zooplankton between inshore and offshore locations has already been reported by other authors (Bode et al., 2003;
Chouvelon *et al.*, 2014) and it could be linked to the different contribution of terrestrial *vs.* marine sources of nitrogen and
carbon moving from inshore to offshore waters, and/or to different trophic dynamics between costal and oceanic food webs.
Here $\delta^{13}C$ values were highly variable in accordance with the wide array of food sources (*i.e.*, marine and continental) available
in the area due to the riverine inputs. Accordingly, the niche width of zooplanktonic community in the area is the greatest and
SEAc decreased southward, where zooplanktonic community were likely sustained mostly by marine sources (Coll et al.,
2007). Standard ellipses were mainly stretched along the x-axis ($\delta^{13}C$) for GSA17N and GSA17C-S showing a progressive
decrease of the continental influence from the Northern to the Central Adriatic basin. SEAc of GSA18 was conversely mostly
extended along the y-axis ($\delta^{15}N$), likely because of the occurrence of a well-structured community with all TLs represent. The
low $\delta^{15}N$ range (and the general high $\delta^{15}N$ values) observed for GSA17N community suggest a shift to omnivory in
zooplanktonic communities in this area to avoid competition (Doi et al. 2010) in high-density condition, as that generated after
the phytoplankton bloom (Bernardi Aubry et al, 2012) here observed in June.

## 5.  Conclusions

This study represents the first application of the stable isotope approach to the analysis of the mesozooplanktonic food web at
Adriatic basin scale including both coastal and offshore communities. The results unveiled the presence of significant
differences in zooplankton abundance, biomass, and community composition at mesoscale level, with the main differences
observed between the Northern Adriatic and the rest of the basin, due to the peculiar oceanographic conditions (i.e., cold
waters) and the strong influence of the Po river. Such differences were also particularly evident in terms of isotopic
composition, where a further separation between offshore and inshore communities were evident for the progressive increase
of marine contribution to food sources for zooplankton in offshore communities. Such findings may represent a valuable
baseline for food web studies encompassing lower to high trophic level species and against changes in oceanographic
conditions under a climate change scenario, considering the rapid response of zooplankton communities to global warming.

**Author contribution**

IL, AdF and SM designed the survey and carried it out. EF conceived the experimental design. EF and SM analysed the
samples. EF analysed the data and prepared the manuscript with contributions from all co-authors.

**Competing interests**

The authors declare that they have no conflict of interest.



**Data availability**
Data can be requested to the corresponding author upon reasonable request.
**Acknowledgements**
The authors wish to thank the crew of the R/V "G. Dallaporta" and all participants of the acoustic survey MEDIAS 2019
GSA 17 and GSA 18.

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
