# Peer review of "Spatial changes in community composition and food web structure of mesozooplankton"

_Biogeosciences, 2021_

## Author Response (AR1)

**Polytechnic University of Marche**

[Figure]

**Department of Life and Environmental Sciences**

Ancona, 23.01.2022

Dear Editor,

Please, find enclosed the edited version of the manuscript by Fanelli et al., entitled: "*Mesoscale variations in the assemblage structure and trophodynamics of mesozooplankton communities of the Adriatic basin (Mediterranean Sea)*", submitted for consideration to Biogeosciences as Original Research Article.

We have carefully followed all the suggestions and comments made by the two anonymous reviewers (hereafter named AR1 and AR2), as highlighted in our first replies to both AR1 and AR2, and made changes to the ms accordingly.

All changes in the manuscript were easily recognizable as we used a red font. With respect to the first replies, after completing the revision, we'd like to add the following comments/replies:

**Reply to reviewer1 (AR1)**

Dear reviewer

On behalf of all co-authors I wish to thank your for the helpful and constructive comments provided, which allow us to greatly improve the manuscript. All your concerns have been addressed and comments/suggestions considered (see below).

Best regards

Emanuela Fanelli

1. The word "mesoscale" in the title and in the objective is misleading, as no evidence is presented on hydrographic features or dynamics (i.e. currents, fronts, eddies,…) characterising the different subareas or were directly related to the described plankton assemblages. As the study conclusions are limited to large subregions, mostly related to the fishery management, the title and the objective should be more precise in this regard (e.g. Differences in taxonomic and trophic mesozooplankton assemblages across the Adriatic basin…).

*Authors: Thanks for the observation, we agree and changed the title accordingly "Spatial differences in community composition and food web structure of mesozooplankton across the Adriatic basin (Mediterranean Sea)"*

AR1

Di.S.V.A.
Via Brecce Bianche
60131 ANCONA
Tel. 0712204747-4331
www.univpm.it

2. The claim that the study is relevant for the whole basin is not entirely supported by the data. The samples were collected on mostly the southern coast, with most of the stations on shallow depths (< 50 m) and none of the stations were located in the deepest waters (>200 m). The study characterized differences between inshore and offshore plankton and also between some subregions. The latter, initially defined from fishery survey areas, were subsequently grouped in the different analysis. However, because the limitations in the sampling, the different zones finally considered for the analysis need to be clarified. In the current version of the manuscript, the initial expectation of having 4 geographic subareas and two major depth categories (Fig. 1) is dismissed in the subsequent analyses. It would be appropriate to define the final analysed regions from the beginning by providing clearly the reasons for grouping the "fishery" areas and use an specific, more intuitive naming (e.g. North, Central, South) instead of the non-intuitive original code names (GSA17N, GSA18,…). Also, the variables used for supporting the groupings must be detailed (e.g. L 129-131), as well as the criteria for the inshore vs. offshore classification (L 176).

*Authors: All the concerns of AR1 were addressed. The sampling design used for statistics were clarified, as well as the selection of inshore and offshore stations. "Fishery" names (i.e., GSA17 and 18) were substituted with geographical names (Northern, central and Southern Adriatic, i.e. NA, CA and SA, according to Artegiani et al., 1997). We agree with the reviewer concerning the lack of specifications in the abstract about the surficial "nature" of our samples and amended this aspect accordingly. The MEDIAS survey effort, the samples came from, is concentrated to the epipelagic zone and thus no samples below 200 m are usually collected (we added this clarification in the abstract and throughout the text). Concerning the spatial effort, this included, samples collected across the entire basin, on the western side, as indicated by haul points in figure 1. In the revised version we also detailed the criteria for the inshore vs. offshore classification, according to the indication provided in Liquete et al., 2011.*

AR1

3. The organization of the manuscript must be improved. There is methodological information among the results (e.g. co linearity tests L 298-300; acidification test L 307-310), excess of auxiliary tables that could be safely moved to supplementary information (e.g. Tables 1 to 3, table 6), results among the discussion (e.g. L 421-424; 427-429), and analysis not well justified (e.g. PERMANOVA on untransformed data in L 174-177 and later on transformed data in L 182, a priori classification in trophic groups in L 209-213 and cluster in

Di.S.V.A.
Via Brecce Bianche
60131 ANCONA
Tel. 0712204747-4331
www.univpm.it

Fig. 4). In addition there are several inaccuracies and inconsistencies in the naming of variables and data through the text and tables (e.g. taxa with unknown trophic group = "Unk" or "?").

*Authors: we wish to thank AR1 for his/her indications about ms organization improvement. We have carefully considered this aspect in preparing the revised version. Auxiliary tables (1-3 and 6) have been moved to Supplementary materials, as requested. Inaccuracies and inconsistencies have been carefully checked and corrected.*

AR1

The abundance and biomass comparisons are made on data averaged in the water column (i.e. N or B / m3). Given the difference in the water column depths sampled in the different stations this approach tend to reduce the importance of offshore stations as the numbers are "diluted" in a large volume of water. In this case a normalization per unit surface (m2) may be appropriate (or at least explored). Also, the use of a logarithmic scale in the relevant plots (e.g. Fig. 2a) will facilitate the comparison of values.

*Authors: we really thank the reviewer for this observation. We have normalized data to surface (instead volume) and changes all the analyses and graphics accordingly (lines 195-198 in M&M and related paragraphs in results and discussion sections). As results changed, we have also rewritten and rediscussed them, in the light of such changes. We have also changed the scale (to logarithmic) in the related plots.*

AR1

4. The isotopic niche comparisons are flawed because of the lack of samples of species of low trophic position (TP<3) in one of the areas (Table 5). It would be more appropriate to limit the comparison of ellipses to all three areas but only for species with TP>=3 and the comparison of species to the ones having samples in all areas (e.g. C. helgolandicus, Chaetognatha, and Decapoda-zoea). In addition, the trophic position estimations are based on only one d15N measurement of G. tenuispinus from the central area that is applied to all areas. The authors must clearly justify the use of this baseline taking into account that the feeding of this species is ranked as "Unknown". The use of a single baseline implies that the average source of N is the same across the basin which does not seem the case because of the inputs of the Po river (e.g. L

Di.S.V.A.
Via Brecce Bianche
60131 ANCONA
Tel. 0712204747-4331
www.univpm.it

34-36, 125-126). Have the authors explored the possibility of using other values reported for zooplankton in the region?

*Authors: We have few more samples available for species belonging to TL2 for the Northern Adriatic (namely NA) sub-area. Oithona samples for NA were processed for SIA and data re-analysed. We have now used three different values as baselines for the three areas, specifically the average values of filter feeder-herbivore taxa, all details have been now provided in the revised version. No isotopic data for zooplankton from the Adriatic Sea have been found, with the only exception of a very local study from the coasts of Croatia published in 2021, where the average values of mesozooplankton (with no indication of the species used and/or their trophic guild) was used to calculate the trophic position of some demersal species (Zorica et al., Front Mar. Sci.* https://doi.org/10.3389/fmars.2021.609432*). Moreover, the use of other values reported for zooplankton in the region would have not been fully appropriate for the possible bias generated by using different IRMS and labs, as highlighted in Mill et al., 2007 (Mill AC, Sweeting CJ, Barnes C, Al-Habsi SH, MacNeil MA, 2008. Mass-spectrometer bias in stable isotope ecology. Limnol Oceanogr Methods 6:34–39).*

**AR1**

In addition there are several minor issues that need the attention of the authors, as exemplified below:

*Authors: All minor issues have been addressed.*

L 21: "DistLM model" is not defined previously and perhaps can be safely removed from the abstract.

*Authors: Spelled-out in the abstract in full*

L 26-28: This sentence is too generic as a conclusion of the study

*Authors: OK, changed with a more specific conclusion.*

L 31-56: This paragraph is too long.

*Authors: OK, divided in sub-paragraphs*

Fig. 1. Explain the "hauls" or remove them (samples not used in this study). Add the final area names (North, Central, South as suggested) along with the "fishery survey codes".

Di.S.V.A.
Via Brecce Bianche
60131 ANCONA
Tel. 0712204747-4331
www.univpm.it

*Authors: OK, removed.*

L 120: explain "acoustic sampling"?

*Authors: OK, done*

L 123: indicate here the variables measured (L 190-192)

*Authors: OK, done*

L 133: fresh weight? indicate precision (L 138)

*Authors: OK, done*

L 166-170: Perhaps is not necessary this detail as the use of a flowmeter is a standard practice in zooplankton sampling (e.g. Harris, R.P., Wiebe, P., Lenz, J., Skojdal, H.R., Huntley, M., 2000. ICES zooplankton methodology manual. Academic Press, San Diego).

*Authors: OK, deleted and referred to Harris et al, 2000*

L 176: define "inshore and offshore"

*Authors: OK done*

L 190-191 pressure units are dbar not "db", chlorophyll fluorescence?, salinity has no units (remove PSU), density units must be kg/m3 not "km/m3"

*Authors: OK corrected*

L 241: The correction of A. clausi (instead of A. tonsa) also affects other parts of the manuscript (L 377, 347)

*Authors: OK corrected throughout the text*

Fig. 2. Station names are almost unreadable; indicate the number of samples for each bar; explain the meaning of box-plot symbols (box, whiskers, line)

*Authors: OK done*

Di.S.V.A.
Via Brecce Bianche
60131 ANCONA
Tel. 0712204747-4331
www.univpm.it

Fig. 3. Consider drawing circles around the samples in the main zones for clarity.

*Authors: This is no longer necessary as the new CAP clearly separates the groups.*

Table 4. Indicate all the variables used in the heading (t, S, DO, fluorescence, turbidity,...?)

*Authors: OK done*

Table 5. In addition to this table consider making accessible the raw isotope data per species in an international repository (e.g. https://www.pangaea.de). These data will be invaluable as further reference for this region.

*Authors: Yes, we considered this suggestion, and we are starting the process of submission to Pangaea.*

Figs. 5 and 6, and Tables 6 and 7. The information provided is misleading of the isotopic differences between stations, as the distribution of species and trophic categories is very different among them (e.g. lack of herbivores in the north area).

*Authors: This part has been completely revised (see comments above), moreover some primary consumers have been analysed for NA.*

L 373-380. Despite the general coincidence with previous studies, the analysis of samples in this study was incomplete (L 243-245). Is there additional information of the formalin-preserved samples (L 122)? Detail the bias caused by the use of frozen samples for community analysis.

*Authors: We agree with reviewer's observations, but the strength of this study in our opinion mostly relay on the spatial coverture of the sampling and on the trophic aspect based on SIA. We already recognized that we missed a part of the species because of the freezing, this was explicated in the text at lines is more This was already explicated "Such differences maybe only apparent and attributable to the storage method we used, as samples were kept frozen for subsequent stable isotope analyses, determining a damage in many organisms, which were impossible to identify to species or even genus level (Fanelli et al., 2011)."*

L 381-382. The difference in mean volumetric abundance may not be the same when integrated values are considered

*Authors: OK, now changed, and discussed*

Di.S.V.A.
Via Brecce Bianche
60131 ANCONA
Tel. 0712204747-4331
www.univpm.it

L 434-444. The match between a priori defined trophic classifications and estimated trophic positions from d15N needs further justification. From the text it is not clear whether the trophic classifications were based on species-specific studies or in inferences from available information on species. In this regard there are several references providing trophic classifications for some zooplankton groups (e.g. http://doi.org/10.1093/plankt/fbv096; https://esajournals.onlinelibrary.wiley.com/doi/abs/10.1890/15-1275.1)

*Authors: OK, this part has been changed also considering the classification suggested in those two papers, really thanks for this.*

L 467-468. However, predation on protozoa may have been overlooked by traditional stable isotope measurements (e.g. https://doi.org/10.4319/lo.2014.59.5.1590)

*Authors: Done, see lines 457-461. Thank you very much for this suggestion.*

L 471-472. The isotopic comparison must consider species (or TPs) present in all areas

*Authors: this aspect has been amended, see reply to comment 4.*

L 473-474. Some more recent similar studies comparing community assemblages and trophic variables: https:// doi.org/ 10.3389/fmars.2018.00498; https://doi.org/10.1016/j.dsr.2020.103402

*Author: Thank you very much for this indication, the paper by Espinosa-Leal and colleagues has been cited.*

**Reply to reviewer2 (AR2)**

Dear reviewer,

we are very grateful for your precious comments which allowed us to better define mesozooplanktonic communities of the Adriatic basin in terms of assemblage structure, mostly liked to the peculiar oceanographic settings of the area. All your comments have been considered and a revised version of the manuscript has been prepared following your recommendations.

Best regards

Emanuela Fanelli

AR2

Di.S.V.A.
Via Brecce Bianche
60131 ANCONA
Tel. 0712204747-4331
www.univpm.it

This paper describes the spatial distribution of mesozooplankton communities in the whole basin of the Adriatic Sea as well as the food structure of mesozooplankton based on isotope analysis. In general, the authors attempted to present the spatial differentiation of mesozooplankton community among the different areas of the Adriatic Sea. Although to be presented the zooplankton community for the whole basin is something rather new for the area, their findings regarding the observed differences in the mesozooplankton community between offshore and inshore among the areas it is something that is expected. What is missing is to investigate if the observed differences can be attributed to different mesoscale features existing in the area. However, the innovative part of this study are the results for the isotope analysis. The authors have used many species and they tried to find through the isotope analysis the food web structure of the mesozooplankton in the Adriatic. Although their findings are quite interesting and contributes with new knowledge especially regarding the findings from the isotope analysis, there are some parts in the methodology as well as in the results that the authors have to clarify before this article has the quality to be published. In the following notes, I will give some comments and suggestions on the different chapters that could help the Authors to improve their article.

*Authors: Thanks, all the concerns have been now addressed.*

In the abstract, a relative to the results conclusion is missing. Please add a more specific conclusion regarding to your findings.

*Authors: Thanks, we have added a sentence in this sense.*

Either in the Introduction or in the Methodology/study area description chapter, it would be helpful the authors to add a description of the hydrological conditions /mesoscale features/circulation that they existing in the Adriatic Sea during the sampling period.

*Authors: Thanks for this suggestion, we have added this missing information (see lines 95-110, 117-119 and 124-126), moreover we have added in the supplementary material two tables with the average values of the environmental variables and the related statistics, which were commented in the results section (see lines 311-323).*

Line 110. It seems that the selected areas are based only to the topograpfy and fishery management reasons. What about the existing circulation and mesoscale features of

Di.S.V.A.
Via Brecce Bianche
60131 ANCONA
Tel. 0712204747-4331
www.univpm.it

the area? Have you take into consideration other parameters as well for the division of the area?

*Authors: The project this paper is based on is finalized to the assessment of small pelagic fishes, this was the reason for our first choice. However, according to your suggestions and considerations from AR1, we have changed this approach and following the division in sub-basin according to oceanographic conditions as reported in Artegiani et al., 1997 and Russo and Artegiani, 1997 (see lines 78-89 and 152-155).*

Line 116. Explain the reasons that you have done the zooplankton sampling in June/July since it is not one of the corresponding period to examine zooplankton in the open sea.

*Author: As reported in the previous reply, the sampling of zooplankton was carried out within the framework of an international survey (MEDIAS) aimed at the assessment, through acoustic methods and ground-truthing, of small pelagic fishes, throughout the whole Mediterranean Sea, encompassing the areas of higher densities for this species. For this reason, the survey is concentrated in the late spring-early summer period, a favorable period for small pelagics' sampling. We acknowledge your observation and add a better rationale for this choice, also in view of further analysis of those samples aiming at assessing the importance of zooplankton for the diet of small pelagic fishes in the basin.*

Line 123-131. Besides the selected oceanographic characteristics to separate the areas, have you tested if also other oceanographic variables such as T, S, O2, Chla and nutrients are significantly differentiated among them?

*Authors: We have now performed statistical tests for each variable, a supplementary table (Table S6) with the results was added in Supplementary Material. We have also added a table (S5) with the mean values of T, S, O2, and Chla, for each sub-area, separating inshore vs. offshore stations.*

In addition, what is the sampling bottom in areas GSA17S and GSA18? In the figures below there are six areas. It seems that the description of the selected areas is rather confusing. Thus, please clarify better how you have done the separation. It would be helpful if there is a Table to present the oceanographic characteristics and variables from the different areas as well as the significant differences among them

*Authors: Thanks, this comment was considered, and table added, see reply to the comment above.*

Line 135-137. Some lines before (121-123) you have mentioned that you separated each sample in two subsamples. The first one is frozen and the second one is preserved in formalin. However, you preferred to do the identification of the

Di.S.V.A.
Via Brecce Bianche
60131 ANCONA
Tel. 0712204747-4331
www.univpm.it

zooplankton community in species level from the frozen samples. Can the authors explain why they have not chosen the preserved samples for their analysis, since the animals in the preserved samples will be in a better condition for their identification?

*Authors: we need the samples for the isotopic analysis, of course we'd have analysed also the portion in formalin, but we cannot then consider this portion for SIA, while doubling our effort. This kind of approach has been used before in other papers (Rumolo P., Fanelli E., Barra M. Bonanno A. et al. Trophic relationships of zooplankton in the Central Mediterranean sea: a stable isotopes approach. Hydrobiologia Special issue, DOI 10.1007/s10750-017-3334-9; Madurell T., Fanelli E., Cartes J. E. (2008). Isotopic composition of carbon and nitrogen of suprabenthos fauna in the NW Balearic Islands (western Mediterranean). J. Mar. Syst. 71, 336-345; Fanelli E., Cartes J.E., Rumolo P., Sprovieri M. (2009) Food web structure and trophodynamics of mesopelagic-suprabenthic deep sea macrofauna of the Algerian basin (Western Mediterranean.) based on stable isotopes of carbon and nitrogen. Deep Sea Research I, 56: 1504-1520); Cartes J.E., Fanelli E., Papiol V., Zucca L. (2010). Distribution and diversity of open-ocean, near-bottom macrozooplankton in the western Mediterranean: analysis at different spatio-temporal scales. Deep Sea Research I 57(11): 1485-1498; Fanelli E., Cartes J.E., Papiol V. (2011) Trophodynamics of zooplankton fauna on the Catalan slope (NW Mediterranean, etc.): insight from $\delta13C$ and $\delta15N$ analysis. Journal of Marine Systems 87: 79-89) and although it is true that not all the specimens can be identified at species level, this allows to also analyse the samples for SIA.*

Line 138-139. Explain why you have chosen to use the wet weight biomass and no other expression of biomass like dry weight or in carbon.

*Authors: as for the previous reply, samples were used for SIA, thus specimens were weighted after the identification, then some specimens were selected for SIA while preserving the other material for potential further analyses.*

Section 2.7. It would be good to compare the isotopic niche using the species found in all sampling areas. Clarify why you have taken Gaetanus as a baseline, since you have found it only in one station. You should clarify why you made such a decision, otherwise it would be good for the baseline to use a common species throughout the study area. Use also recent publications for the eastern Mediterranean for your comparison and discussion.

*Authors: Many thanks for this comment, according also to a similar comment from AR1, we have completely restructured this part considering the average values of filter feeders-herbivores from each area as baseline for the three different sub-areas. The recent work of Protopapa et al. (2019) for the Eastern Mediterranean has been also considered (both in M&M and in discussion).*

Di.S.V.A.
Via Brecce Bianche
60131 ANCONA
Tel. 0712204747-4331
www.univpm.it

[Figure]

Line 251-253. Do you mean here that there are not inshore-offshore differences between the same areas or between the different subareas? Please clarify your results. It would be more helpful to show in the Table the significant differences or not among the selected areas. It is difficult to follow the results.

*Authors: Thanks, we have amended the existing table, to better clarify our results, sorry for this.*

Also the observed significant differences are only for the South to North areas? What about the central one?

*Authors: Thanks, as mentioned we have changed this part in the light of the new results.*

Line 274-275. There is also a clear separation of GSA18 offshore from the other stations and the offshore stations of the GSA17N is close to those of the GSA17C-S according to the figure. Please explain better the results.

*Authors: the new CAP showed a clear separation among the three sub-areas. This part has been re-written.*

Line 278-279. In the Figure 3 legend, what is the sub area 1??

*Authors: Sorry, it was an oversight, this is the name given to the factor, now corrected in "sub-area".*

Section 4.2. Almost the whole paragraph should be moved to the results section in order to understand why you have made the selection of the areas.

*Authors: This paragraph has been re-arranged and part moved to results' section, thanks.*

Line 431-432. DO is not only the major factor according to your analysis. Salinity and fluorescence have higher percentage. What about them?

*Authors: According the new results, we have changed this part.*

Di.S.V.A.
Via Brecce Bianche
60131 ANCONA
Tel. 0712204747-4331
www.univpm.it

Emanuela Fanelli, on behalf of all co-authors

[Figure]

--<°)))><--- <°)))><--- <°)))>< ---<°)))><
Emanuela Fanelli, PhD
Associate Professor (BIO/O7)
Lab of Marine Biology and Ecology
Dept. of Life and Environmental Sciences
Polytechnic University of Marche , Ancona (Italy)
Webpage:https://www.univpm.it/Entra/Scienze_1/docname/idsel/775/docname/EMANUELA%20FANELLI
Scholar: https://scholar.google.it/citations?user=zdXIr1AAAAAJ&hl=it
--<°)))><--- <°)))><--- <°)))><--- <°)))><

Di.S.V.A.
Via Brecce Bianche
60131 ANCONA
Tel. 0712204747-4331
www.univpm.it